# Analysis of Characteristics of Bovine-Derived Non-Enterotoxigenic *Bacteroides fragilis* and Validation of Potential Probiotic Effects

**DOI:** 10.3390/microorganisms12112319

**Published:** 2024-11-14

**Authors:** Dong Wang, Long Zhao, Jingyi Lin, Yajing Wang, Haihui Gao, Wenhui Liu, Qirui Li, Liang Zhang, Xiaodong Kang, Kangkang Guo

**Affiliations:** 1College of Veterinary Medicine, Northwest A&F University, Xianyang 712100, China; blackwhiteelegant@163.com (D.W.); zhaolong4480@sina.com (L.Z.); ljy18309788119@163.com (J.L.); dabiaqu@163.com (Y.W.); gao6600022@163.com (H.G.); wenhui9953016@sina.com (W.L.); lqr365698@163.com (Q.L.); eerduosizl@163.com (L.Z.); 2Institute of Animal Science, Ningxia Academy of Agriculture and Forestry Sciences, Yinchuan 750002, China

**Keywords:** bovine-derived, *Bacteroides fragilis*, intestinal health, prediction of strain characteristics, prevention of diarrhea

## Abstract

*Bacteroides fragilis* is a new generation of probiotics, and its probiotic effects on humans and some animals have been verified. However, research on *B. fragilis* in cattle is still lacking. In this study, 24 stool samples were collected from two large-scale cattle farms in Wuzhong, Ningxia, including 12 diarrheal and 12 normal stools. A non-toxigenic *Bacteroides fragilis* (NTBF) was isolated and identified by 16S rRNA high-throughput sequencing and named BF-1153; genome composition and genome functional analyses were carried out to reflect the biological characteristics of the BF-1153 strain. A cluster analysis of BF-1153 was performed using Mega X to explore its genetic relationship. In addition, Cell Counting Kit-8 (CCK8) was used to determine the toxic effects of the strain on human ileocecal colorectal adenocarcinoma cell line cells (HCT-8), Madin-Darby bovine kidney cells (MDBK), and intestinal porcine epithelial cells (IPECs). The results showed that BF-1153 conformed to the biological characteristics of *B. fragilis*. BF-1153 had no toxic effects on HCT-8, MDBK, and IPEC. Animal experiments have shown that BF-1153 has no toxic effects on healthy SPF Kunming mice. Notably, the supernatant of BF-1153 enhanced cell activity and promoted cell growth in all three cell lines. At the same time, a cluster analysis of the isolated strains showed that the BF-1153 strain belonged to the same branch as the *B. fragilis* strain 23212, and *B. fragilis* strain 22998. The results of the animal experiments showed that BF-1153 had a certain preventive effect on diarrhea symptoms in SPF Kunming mice caused by a bovine rotavirus (BRV). In summary, the strain BF-1153 isolated in this experiment is NTBF, which has no toxic effect on MDBK, HCT-8, and IPEC, and has obvious cell growth-promoting effects, especially on MDBK. BF-1153 promotes the growth and development of SPF Kunming mice when compared with the control group. At the same time, BF-1153 alleviated the diarrhea symptoms caused by BRV in SPF Kunming mice. Therefore, BF-1153 has the potential to be a probiotic for cattle.

## 1. Introduction

Diarrhea is one of the most common safety concerns worldwide; it is the third leading cause of death in children under 5 years old, causing about 443,832 deaths per year, and seriously affecting both people and animals. Diarrhea is defined as the discharge of loose stools or liquid stools three or more times per day [1]. This problem is particularly severe in juvenile animals. Diarrhea is a clinical manifestation caused by damage to ion channels, ion transport channel proteins, and physicochemical barriers in the intestines due to external factors (pathogens, environment) and is accompanied by disturbances in water and electrolyte transport in the digestive tract [2,3]. An animal intestine contains a large number of different kinds of microorganisms, and with a change in a gastrointestinal segment, the species and abundance of microorganisms also changes. Therefore, the disturbance of water and electrolytes in the gastrointestinal tract destroys the dynamic balance of the intestinal microbiota, which in turn leads to the destruction of the intestinal barrier, increases susceptibility to other pathogens, and causes many diseases, such as enteritis, irritable bowel syndrome (IBS), allergies, cardiovascular diseases, and obesity [4,5,6]. Therefore, the dynamic balance of gut microbiota is the basis for peaceful coexistence between hosts. Through metagenomic studies, Eckburg found that the phylogenetic status of gut microorganisms can be roughly classified into *Firmicutes*, *Bacteroidetes*, *Proteobacteria*, *Actinobacteria*, *Verrucomicrobia*, *Fusobacteria*, and six other species, and *Bacteroides* and *Firmicutes* are the main dominant flora in the intestine [7]. The dominant flora colonizes specific parts of the intestinal mucosa and multiplies until a stable flora is formed. When the intestinal position is occupied by a dominant flora, pathogenic microorganisms are excluded due to competition. Therefore, *Bacteroides*, as the dominant flora, play an important role in the composition of intestinal flora.

*B. fragilis* is known as a new generation of probiotics [8,9,10,11], while the World Health Organization (WHO) and Food and Agriculture Organization of the United Nations (FAO) define probiotics as “live microorganisms which when administered in adequate amounts confer a health benefit on the host” [12]. Therefore, whether *B. fragilis* meets the definition of probiotics is the main problem to be addressed. Previous studies have shown that enterotoxigenic *Bacteroides fragilis* (ETBF) can infect the abdominal cavity and cause a variety of diseases, such as enteritis, abdominal abscess, soft tissue infection, bacteremia, and colorectal cancer [13,14,15]. In contrast to ETBF, a growing body of research proves that polysaccharide A (PSA) produced by NTBF exerts beneficial effects, with *B. fragilis* PSA as an archetypical bacterial capsular polysaccharide example, directing microbiota and host interactions [16]. PSA is unique because it possesses zwitterionic motifs that enrich a specific subset of anti-inflammatory memory CD4^+^Foxp3 T cells after direct interactions with antigen-presenting cells (APCs), such as plasmacytoid dendritic cells. This can result in a systematic amelioration of inflammation-related diseases, such as abscesses, neuroinflammation, and cancers [17].

Studies on the probiotic effects of an NTBF of human origin [18,19,20] and chicken origin [21] have been reported, but there are relatively few studies on NTBF in calves. In order to screen the key intestinal microorganisms of bovine-derived *B. fragilis*, explore the biological characteristics of bovine-derived *B. fragilis*, and increase the research data for the development of a new generation of probiotics, in this study, we collected diarrheal stool samples and normal stool samples from the Wuzhong, Ningxia Hui Autonomous Region (Ningxia, China). Using 16S rRNA high-throughput sequencing, we determined the microbial composition of diarrheal and normal stools and explored the differences in intestinal microorganisms between diarrheal calves and normal calves. Finally, we isolated and cultured the key intestinal microorganisms and verified their biological characteristics.

## 2. Materials and Methods

### 2.1. Sampling and 16S High-Throughput Sequencing

In the Shangle farm and Wuli farm of the Ningxia Hui Autonomous Region (Ningxia), sterile EP gloves were used for calf rectal sampling, and 12 normal calf rectal stools and 12 diarrheal calf rectal stools were used. The details of the samples are presented in Appendix A.

After sending the collected diarrheal stool samples and normal stool samples to the laboratory at 4 °C, the 200 g fecal samples were aliquoted into multiple 2 mL sterile EP tubes in a sterile bench, then frozen in liquid nitrogen for 5 min, wrapped in a foam box with dry ice, and sent to Tsingke Biotechnology (Beijing, China) for 16S high-throughput sequencing to determine the key microorganisms in the gut.

### 2.2. Isolation and Identification of Bacteroides fragilis

Normal stool samples were picked and inoculated on a sterile station into the bottom of a tube containing 5 mL of an autoclaved Brain Heart Infusion broth (BHI) medium (Hope Bio-Technology, Qingdao, China) and sealed with 1–2 mL of sterile liquid paraffin on top of the medium. After shaking the bacteria at 37 °C and 120 r/min for 1–2 d at a constant temperature, the bacterial solution was inoculated onto the plates of a solid bacteroides bile esculin (BBE) agar medium (Hope Bio-Technology, Qingdao, China) with sterile applicator sticks, and the coated plates were placed into 2.5 L anaerobic culture bags (Hope Bio-Technology, Qingdao, China), then 2.5 L anaerobic gas-producing bags (Hope Bio-Technology, Qingdao, China) were added, the plates were incubated at a constant temperature of 37 °C for 2~3 d, and then the growth of the plates was observed.

The single colonies were selected, inoculated into 15 mL of a BHI sterile broth, 1–2 mL of a sterile liquid paraffin was added for sealing, and the OD_450_ value was determined by aspirating 200 μL of an uncultured bacterial solution, and then the anaerobic culture was aspirated at 37 °C and 120 r/min to determine the OD_450_ value regularly. Finally, the bacterial growth curve was obtained. After extracting the DNA from the cloudy bacterial solution, the bacteria were identified by PCR using the *B. fragilis* specific primers LEU. ETBF secretes a 20 kDa zinc-dependent metalloprotease toxin (BFT), whereas NTBF cannot secrete BFT. PCR identification was performed using BFT primers and primers for the three isomers of BFT, namely, BFT-1, BFT-2, and BFT-3. The amplification program of the LEU gene consisted of 1 cycle at 95 °C for 3 min, followed by 34 cycles at 95 °C for 30 s, 54 °C for 30 s, and 72 °C for 30 s. The amplification program for the BFT gene consisted of 1 cycle at 94 °C for 4 min, followed by 35 cycles at 94 °C for 15 s, 66 °C for 30 s, and 72 °C for 30 s. The amplification program for BFT-1, BFT-2, and BFT-3 consisted of 1 cycle at 94 °C for 4 min, followed by 35 cycles at 94 °C for 15 s, 60 °C for 30 s, and 72 °C for 30 s. The primers used are listed in Table 1.

The DNA from the BF-1153 strain, which was positive as determined by PCR using the *B. fragilis* specific primers, was extracted, and the 16S rRNA universal primers Eubac27 F/Eubac1492 R were used for PCR amplification and sent to Tsingke Biotechnology (Beijing, China) for sequencing. The sequencing results were compared with the l6s rRNA sequences in GenBank and used to build a phylogenetic tree. The primer sequences used are listed in Table 1.

The genome sequence was annotated using Prokka (v1.14.6) software to obtain the gff file, and the gff file was processed using roary (v3.13.0) to obtain the core gene multiple sequence alignment file (Core_Gene_Alignment.fasta). ModelFinder v2.2.0 was also used to find the best evolutionary tree model (GTR+F+I+I+R8), and finally a phylogenetic evolutionary tree based on the core genome was constructed using iq-tree (v2.3.6, maximum likelihood method, ultrabootstrap 1000, model: GTR+F+I+I+R8).

### 2.3. Whole-Genome Sequencing (WGS) Analysis of BF-1153 Strain

#### 2.3.1. DNA Extraction and Sequencing

The total DNA of the bacteria was extracted using the OMEGA genome kit. After the DNA was qualified by electrophoresis, it was sent to Xi’an Tsingke Biotechnology for library construction. After the DNA fragments were processed, the library was completed after sequencing steps such as end repair, A tail addition, and purification. After the library inspection was qualified, the different libraries were sequenced on the Illumina HiSeq platform after pooling according to the effective concentration and the target offline data volume. After pretreatment, Clean data were obtained, and assembled using SOAP denovo (version 2.04), SPAdes (version 3.15.5), ABySS (version 2.0) assembly software. Finally, CISA software (http://sb.nhri.org.tw/CISA, accessed on 22 November 2022) was used for integration, and then gapclose (version: 1.12) and other software were used to optimize the preliminary assembly results to obtain the final assembly results. Finally, the fragments below 500 bp were filtered out, evaluated, and statistically analyzed, and a subsequent gene prediction was performed.

#### 2.3.2. BF-1153 Genomic Composition Analysis

The sequenced BF-1153 strain was subjected to a GC-Depth analysis by sequencing, genome repeats, and non-coding RNA (ncRNA) were detected, and the coding genes, gene islands, prophages, and Clustered Regularly Interspaced Short Palindromic Repeat Sequences (CRISPR) were predicted. The GC bias of the strain genome and the proportion of repeated sequences contained can be obtained, and the species composition specificity of the repeated sequences can be analyzed. The number and average length of the coding genes of the strains were determined to show the distribution of the genome. At the same time, the ncRNA information and gene island distribution of the sequenced strains were predicted, and the biological characteristics and protein functions of the strains were verified. In addition, it can also evaluate the content of prophages and CRISPR sequences, and obtain genetic sequence information such as antibiotic resistance, environmental adaptability, and the pathogenicity of strains.

### 2.4. Functional Annotation of the of BF-1153 Genome

The sequencing company BLAST software (version 2.14.0) (blastp, default parameter, e value < 1 × 10^−5^) was used in gene ontology (GO), the Kyoto encyclopedia of genes and genomes (KEGGs), and the cluster of orthologous groups of proteins (COGs). The pathogen host interactions database (PHI), virulence factors of pathogenic bacteria (VFDB), and antibiotic resistance genes database (ARDB) were compared with the genome information of the sequenced strains.

### 2.5. Phenotypic Validation of Resistance of Bovine-Derived BF-1153

According to the results of the functional annotation obtained by ARDB and CARD, BF-1153 monoclonal inoculation was selected and placed into 5 mL of a BHI sterile medium, an anaerobic culture was carried out for 24–48 h. The turbidity of the bacterial solution was adjusted to 0.5 McF with sterile normal saline, the adjusted bacterial solution was evenly coated on the BBE solid medium with a sterile coating rod on the ultra-clean table, and then the antimicrobial susceptibility paper was pasted on the surface of the agar, and an anaerobic culture was carried out for 24–48 h. The size of the inhibition zone was observed, and the diameter of the inhibition zone was determined to verify whether the resistance genes of BF-1153 are consistent with the resistance phenotype. The selected antibiotics were lincomycin, erythromycin, azithromycin, tetracycline, doxycycline, trimethoprim.

### 2.6. Investigation of Toxic Effects of Bovine-Derived BF-1153

BF-1153 was streaked and inoculated on a BBE agar. After an anaerobic culture at 37 °C for 1–2 d, a single black colony was picked with a sterile inoculation ring and placed in an EP tube containing 15 mL of sterile BHI. The medium was sealed with 1–2 mL of sterile liquid paraffin and cultured at 37 °C and 200 r/min. After the bacterial solution was turbid, it was filtered on an ultra-clean bench using a 0.22 μm filter, and the filtered bacterial solution was diluted (5^−1^–5^−7^).

Each of the three cell lines (HCT-8: ATCC CCL-244; MDBK: ATCC CCL-22; and IPEC: ATCC GT2759C) (Purchased from ATCC, preserved by our research team laboratory) were seeded in 96-well plates. When the cell density reached approximately 80%, the cells were washed three times with PBS, and 100 μL of the diluted bacterial supernatant was inoculated into the corresponding 96-well plates. At the same time, the sterile BHI control group (5^−1^&2^−1^) and a 1.5% fetal calf serum (FBS) Dulbecco’s modified eagle medium (DMEM) were used as control groups. After 60 h of incubation, the liquid in the hole was discarded and 10 μL of the CCK8 Cell Proliferation and Toxicity Detection Kit was added. The OD_450_ values at 30 min, 60 min, 90 min, 120 min, 180 min, and 240 min were measured.

### 2.7. Animal Experiment of BRV-Induced Diarrhea in SPF Kunming Mice

Twenty healthy SPF Kunming mice of approximately 10 d old were selected and divided into four groups on average. The diarrhea model was established using BRV with a TCID_50_ of 10^−3.5^/100 μL. The experimental group was intragastrically administered 25 μL, 50 μL, and 100 μL of the BRV virus solution daily for 2 consecutive days, and the control group was intragastrically administered 100 μL of the sterile DMEM daily for 2 consecutive days. The changes in stool shape in the mice were observed daily. During the experiment, the mice in each group were fed a normal diet.

### 2.8. Identification of the Toxic Effects of BF-1153 in SPF Kunming Mice

The isolated BF-1153 bacterial liquid was inoculated on the BBE agar, and the colony-forming unit (CFU) of the BF-1153 bacterial liquid was determined after 1–2 d of an anaerobic culture at 37 °C. Ten healthy SPF Kunming mice, approximately 21 d old, were selected and divided equally into two groups. Then, a 1 mL disposable syringe was used to absorb the BF-1153 bacterial solution after the CFU determination, and the SPF Kunming mice in the experimental group were intragastrically administered for 6 d. The mice were fed normally, and body weight changes were observed daily. The same dose of sterile saline was used in the control group and all the other procedures were the same.

Twenty healthy SPF Kunming mice about 8 days old were selected and divided into four groups on average. A disposable syringe was used to absorb 250 μL, 500 μL, and 1 mL of the BF-1153 bacterial solution after CFU determinations were used for gavage, and the mice were fed normally during the test to observe whether the mice had diarrhea. The control group of SPF Kunming mice were treated with the same dose of sterile normal saline, and other the operations were the same.

### 2.9. Protective Effect of BF-1153 on BRV-Induced Diarrhea in SPF Kunming Mice

Fifteen healthy SPF Kunming mice aged approximately 10 d were selected and divided into three groups on average. The first group was intragastrically administered BRV with a TCID_50_ of 10^−3.5^/100 μL for 2 consecutive days, and then the small intestine tissues of the infected mice were collected and fixed with 4% paraformaldehyde for HE staining. Intestinal changes were observed in the mice. The second group was intragastrically administered 100 μL of the *B. fragilis* strain for a CFU determination, and then intragastrically administered BRV with a TCID_50_ of 10^−3.5^/100 μL for 2 consecutive days. The small intestine tissues of the infected mice were collected and fixed with 4% paraformaldehyde for HE staining. Intestinal changes were also observed in the mice. The control group was intragastrically administered 100 μL of sterile saline, and the mice in each group were fed a normal diet during the experiment.

### 2.10. Statistical Analysis

The statistical analyses were performed using GraphPad version 9.0.0. Chi-square tests were performed at the 5% level of significance.

## 3. Results

### 3.1. Microbial Population Abundance in Normal Stool and Diarrhea Stool

The microbial diversity of the stools was analyzed by 16S rRNA high-throughput sequencing, and the sequences were obtained. According to the sequence composition, the samples were subjected to a taxonomic analysis at each taxonomic level, and a community structure diagram of the stool samples from the Shangle farm at the genus taxonomic level was obtained. The results showed that the most abundant bacterial species in each stool from the Shangle farm was *Bacteroides* (Figure 1A,B), and the average content of the *Bacteroides* in the normal stools was significantly higher than that in the diarrhea stools (Figure 1C). The same method was used to obtain the community structure map of the stool samples at the genus taxonomic level on the Wuli farm. The results showed that the most abundant bacterial species in each stool from the Wuli farm was *Bacteroides* (Figure 1E,F), and the average content of the *Bacteroides* in the normal stools was significantly higher than that in the diarrhea stools (Figure 1G).

*Cryptosporidium* was the only pathogen detected in all normal calf stool samples (No. 120, No. 123, No. 126, No. 127, No. 129, No. 131, No. 237, No. 238, No. 239, No. 240, No. 241, and No. 242) from the Shangle farm and Wuli farm, whereas *Cryptosporidium*, *Escherichia coli* K99 (*E. coli* K99), *Giardia*, bovine astrovirus (BoAstV), and bovine coronavirus (BCoV) were detected in the diarrheal calf stool samples (No. 121, No. 122, No. 124, No. 125, No. 128, No. 130, No. 224, No. 225, No. 227, No. 228, No. 233, and No. 235). As mentioned above, the abundance of *B. fragilis* in the stool samples of the normal calves was higher than that in the diarrheal calves. Therefore, in the samples collected in this study, the abundance of *B. fragilis* was inversely proportional to the number of pathogens in the calf stools; that is, the higher the abundance of *B. fragilis* in the normal stools and the fewer pathogen species present, the lower the abundance of *B. fragilis* in the diarrheal stools and the more pathogen species present. The detailed results are presented in Table 2 and Figure 1D,H.

### 3.2. Growth Characteristics of Bovine-Derived Bacteroides fragilis

After 12 normal stool samples were inoculated with the BHI medium for 12 h, the bacterial solution became turbid, indicating the growth of the bacterial solution. The bacterial solution coated on the specific BBE agar medium was cultured under anaerobic conditions at 37 °C. After 1–2 d, all 12 samples grew specific black round single colonies, and the surface was smooth and opaque, which was consistent with the growth of *B. fragilis* on the BBE agar. Single black colonies were selected for strain identification, and only No. 131 was successfully sequenced and identified as *B. fragilis*, named BF-1153. The detailed results are shown in Figure 2A,B. The growth curves of the strains are shown in Figure 2C.

Single colonies were selected and inoculated again into 5 mL of a BHI-containing sterile medium. After the bacteria were cultured under anaerobic conditions until they were turbid, the *B. fragilis* specific primers LEU and enterotoxin gene BFT were used for PCR identification of the bacteria. The LEU gene showed a band at 135 bp, and the BFT gene and other genes had no bands. Finally, the *B. fragilis* strain was isolated from the Shangle farm and named BF-1153. The detailed results are illustrated in Figure 2D–F.

### 3.3. The Attribution of Bovine-Derived BF-1153 and the Construction of a Phylogenetic Tree

The 16S rRNA gene sequence was amplified by PCR to obtain a target fragment of approximately 1500 bp. After sequencing, the strain was identified as *B. fragilis*. The sequences were uploaded to GenBank, and BLAST homology comparisons were performed. The strains with the highest homology were selected for multiple sequence alignments. The maximum likelihood value method in Mega X was used to construct the phylogenetic tree. The results showed that the homology of BF-1153, the *B. fragilis* strain NTCC9343, the *B. fragilis* strain ATCC 25285, the *B. fragilis* strain DSM 2151, the *B. fragilis* strain JCM 11019, et al., was as high as 99.51%, as shown in Table 3. The cluster analysis of the isolated strains showed that the BF-1153 strain belonged to the same branch as the *B. fragilis* strain 23212 and the *B. fragilis* strain 22998, as shown in Figure 2G.

The phylogenetic tree based on the core genome showed that BF-1153 and the *B. fragilis* strain HCK-B3 were in the same branch, which was far from the *B. fragilis* NCTC 9343, and the core genome homology was only 49%, as shown in Figure 2H.

### 3.4. Genomic Profile of Bovine-Derived BF-1153

The total length of the BF-1153 gene sequence was 5,469,519 bp, with one contig; with contig N50, it was 274,500 bp; with contig N90, it was 106,978 bp; and the GC content was 43.42%, as shown in Appendix A. At the same time, the genome of the BF-1153 strain was compared with the *B. fragilis* strain NTCC9343 in GenBank, and the results showed that BF-1153 did not contain the enterotoxin gene.

The size of BF-1153 was 5.71 Mb according to K-mer statistics, and the sequencing data were analyzed using a 15-mer. The corrected genome size was 5.56 Mb, the genome heterozygous rate was 0.04%, and the genome repeat rate was 17.26%. As shown in Appendix A, the gene distribution map after the K-mer statistics is shown in Figure 2I. Through the sequencing depth analysis and GC-Depth analysis, it was determined that there were no contaminated sequences or plasmids in BF-1153, and the number of repeat sequences in the target strain was not high, which had a certain specificity. The results of these analyses are shown in Figure 2J.

### 3.5. Component Analysis of Bovine-Derived BF-1153 Genome

A component analysis of BF-1153 includes coding genes, repeated sequences, non-coding RNA, genomics islands, prophages, and a prediction by CRISPR.

The coding genes prediction results showed that the genome size was 5,469,519 bp, the gene number was 4730, the gene average length was 1022 bp, the gene total length was 4,833,822 bp, and the coding genes accounted for 88.38% of the total genome. The statistical information of the predicted results of the coding genes is shown in Appendix A, and the statistical results of the gene length are shown in Figure 2K. Repeat masker (version open-4.0.5) and tandem repeats finder (TRF) (version 4.07b) were used to search for tandem repeats in the DNA sequences. The repeated sequences results showed that there were 182 long terminal repeat (LTR) types, with a total length of 11,036 bp and an average length of 61 bp, accounting for 0.2018% of the total genome. There were 160 tandem repeat sequences, with a repeat size of 7~207 bp and a total length of 9024 bp, accounting for 0.1650% of the total genome. The forecast results are presented in Appendix A, respectively. The non-coding RNA prediction results showed that the ncRNA contained 71 tRNAs, with a total length of 5791 bp and an average length of 77 bp. This was followed by 5S (Denovo) with seven in number, a total length of 770 bp, and an average length of 110 bp but no sRNA. The detailed results of the ncRNAs are shown in Appendix A. The predicted results of the genomic islands are shown in Figure 2L. Prophages on the sample genome were predicted using phiSpy software (version 2.3). The prediction results showed that BF-1153 contained 12 prophages, with a total length of 612,825 bp, and an average length of 51068.8 bp. The detailed results are presented in Appendix A. CRISPRdigger (version 1.0) was used to predict the genome of BF-1153. The predicted results showed that BF-1153 contains three CRISPR numbers, with a total length of 1880 bp and an average length of 627 bp. The detailed results are presented in Appendix A.

### 3.6. Characteristic Function of Bovine-Derived BF-1153

#### 3.6.1. GO Functional Annotation

In the strain BF-1153, 12,130 genes were annotated in the GO database. The annotation results included three categories: red, biological processes; green, cellular components; and blue, molecular functions. Functional annotation information for the genome is shown in Figure 3A.

A total of 5963 genes in 10 categories were annotated in the biological process category, among which more independent genes were involved in the metabolic process (1603), cellular process (1496), and localization (522). A total of 12 categories of 2557 genes were annotated in the cellular component category, among which cell (988), cell part (98), and organelle (194) accounted for a large proportion. A total of 3610 genes in 10 categories were annotated for molecular function, and more independent genes were involved in catalytic activity (1493), binding (1319), and molecular transducer activity (242).

#### 3.6.2. KEGG Functional Annotation

A total of 1915 genes in the strain BF-1153 were annotated in the KEGG database. As shown in Figure 3B, there were six major metabolic pathways, including 1460 genes in 12 categories of metabolism, 169 genes in 4 categories of genetic information processing, 92 genes in 3 categories of cellular processes, and 85 genes in 2 categories of environmental information processing. Human diseases accounted for 11 classes of 70 genes, and organismal systems accounted for 7 classes of 39 genes. They mainly participated in cellular community prokaryotes (42 kos), membrane transport (46 kos), translation (74 kos), drug resistance (antimicrobial (24 kos), the global and overview maps (553 kos), the endocrine system (24 kos), and other metabolic processes.

#### 3.6.3. COG Functional Annotation

By comparing the gene information of the strain BF-1153 with the COG database, 2865 genes were annotated, with a total of 24 categories, as shown in Figure 3C. Among them, cell wall/membrane/envelope biogenesis, carbohydrate transport and metabolism, translation/ribosomal structure, and biogenesis had the largest number of genes (337, 230, and 214, respectively), accounting for 11.76%, 8.03%, and 7.47% of the annotated genes.

The classifications with a high proportion of other functional genes and greater than 5% were energy production and conversion (146, 5.09%), amino acid transport and metabolism (187, 6.53%), coenzyme transport and metabolism (166, 5.79%), transcription (191, 6.67%), replication, recombination and repair (180, 6.28%), inorganic ion transport and metabolism (182, 6.35%), general function prediction only (184, 6.42%), and signal transduction mechanisms (183, 6.38%). The amounts for the cytoskeleton (3, 0.11%), the extracellular structures (8, 0.28%), secondary metabolite biosynthesis, transport, and catabolism (17, 0.59%) were less common.

#### 3.6.4. PHI Functional Annotation

In pathogen host interactions, PHI Phenotype classification includes reduced virulence (100), pathogenicity (39), increased virulence (hypervirulence) (16), loss of pathogenicity (10), lethality (4), a chemical target: sensitivity to chemical (1), and an effector (a plant virulence determinant) (1). Among them, the most common types of mutations are reduced virulence, indicating that the pathogenicity gene of the strain is less than that of the ordinary strain; pathogenicity, representing the number of genes not affected by the strain; increased virulence (hypervirulence), indicating the number of mutated genes in the strain is more than in the normal strain; loss of pathogenicity, representing the number of genes missing in this strain compared to other strains; and lethal, representing the number of mutant genes that cause individual death. The results showed that the mutation number of BF-1153 in relation to reduced virulence was significantly higher than that of the other mutation types. The greater the number of reduced virulence mutations, the weaker the pathogenicity of the strain compared to that of other *B. fragilis* strains, as shown in Figure 3D.

#### 3.6.5. VFDB Functional Annotation

Using Diamond (version 4.6.8) software, the amino acid sequences of the target species were compared with the VFDB database, and the genes of the target species and their corresponding virulence factor functional annotation information were combined to obtain the annotation results. The details are presented in Table 4.

#### 3.6.6. ARDB and CARD Functional Annotation

Using Diamond (version 4.6.8) software, the amino acid sequences of the target species were compared with those in the ARDB database, and the genes of the target species and their corresponding drug resistance function annotation information were combined to obtain annotation results. The detailed results are presented in Table 5.

The resistance gene identifier (RGI) software (https://github.com/karynkomatsu/rgi, accessed on 22 November 2022) provided by the CARD database was used to compare the amino acid sequences of the target species with the CARD database (RGI built-in blastp, default value ≤ 1 × 10^−30^). The resistance gene information annotated in the database was counted according to the comparison results of the RGI. The detailed results are presented in Table 5.

### 3.7. Phenotypic Validation of Resistance of Bovine-Derived BF-1153

The results showed that the resistance phenotypes of lincomycin (lincosamide), erythromycin (macrolide), azithromycin (macrolide), tetracycline (tetracycline), and doxycycline (tetracycline) were all resistant, which was consistent with the predicted resistance genes in the ARDB and CARD annotations. The results are listed in Table 5 and Table 6.

### 3.8. Animal Experiment of BRV-Induced Diarrhea in SPF Kunming Mice

The stools of the SPF Kunming mice in the control group treated with 100 μL of sterile DMEM were normal, brown-shaped stools without diarrhea symptoms. Compared with the control group, the stools of the SPF Kunming mice treated with 25 μL of the BRV virus liquid were brownish yellow, molding soft stools, without obvious diarrhea symptoms. The stools of the SPF Kunming mice treated with 50 μL of the BRV virus liquid were yellow, unformed, soft stools, showing a trend of diarrhea. The stools of the SPF Kunming mice treated with 100 μL of the BRV virus liquid were yellow, unformed, watery stools, with obvious diarrhea symptoms. The degree of diarrhea was the most serious 48 h–72 h after the challenge, and this was accompanied by weight loss, slow movement, and a loss of appetite. The results are shown in Figure 3E.

### 3.9. Investigation of Toxic Effects of Bovine-Derived BF-1153 on Cell Lines

After incubating 100 μL of the filtered BF-1153 bacterial supernatant on HCT-8 for 60 h, BF-1153 showed no toxic effects on the HCT-8 cells at any time point. The cell activity of HCT-8 reached the highest (132%~152%) at all periods (30 min, 60 min, 90 min, 120 min, 180 min, and 240 min) when the concentration of the bacterial supernatant was 5^−1^. The supernatant of the bovine-derived BF-1153 bacterial solution also had a positive effect on the cell activity of HCT-8 at other concentrations, as shown in Figure 4A1. Then, the toxic effects of the bovine-derived BF-1153 supernatant on MDBK and IPEC were determined by the same method; the results showed that the bovine-derived BF-1153 bacterial supernatant had no toxic effect on MDBK, the optimal cell growth concentration was 5^−3^, and the growth activity of the MDBK cells reached more than 200%. The specific results are shown in Figure 4A2. The bovine-derived BF-1153 bacterial supernatant also had no toxic effect on IPEC, and there was no optimal cell growth concentration, so the growth activity of the IPEC cells reached more than 100%. The specific results are shown in Figure 4A3.

### 3.10. Investigation of Toxic Effects of Bovine-Derived BF-1153 on SPF Kunming Mice

The concentration of the bovine-derived BF-1153 bacterial solution was determined by the colony counting method, and the number of viable BF-1153 bacteria in 1 mL was determined to be 1.5 × 10^8^ CFU/mL. On day 0, the average weight of the mice in the control group was 24.79 g. There was no significant difference in body weight between the experimental and control groups. The mice in the experimental group were intragastrically administered 1 mL of the bovine-derived BF-1153 bacteria solution for 6 d, and the mice in the control group were intragastrically administered the same amount of sterile normal saline. The results showed that mice in the experimental group grew normally, and there was no significant difference in weight gain between the experimental and control groups. The average weight gain of the mice in the experimental group was higher than that of the control group. The weight gain rates of the mice in the experimental group were 19.04% and the weight gain rate of the mice in the control group was 14.32%. The weight gain rate of the experimental group was 4.72% higher than that of the control group, indicating that bovine-derived BF-1153 bacterial solution not only had no toxic effect on the SPF Kunming mice but also had a certain role in promoting their growth and development. The results are shown in Figure 4B.

Secondly, after the intragastric administration of different doses of the BF-1153 bacteria solution, the feces of the SPF Kunming mice in the experimental group appeared normal, brown, and solid, with no diarrhea symptoms, which was consistent with the fecal traits of the mice in the control group. The results are shown in Figure 4C.

### 3.11. Bovine-Derived BF-1153 Has Potential Probiotic Effects on SPF Kunming Mice

The surface of the intestine of the normal mice (control group) was a single layer of columnar epithelium, the mucosal epithelial cells were dense in the cytoplasm, the cells were tightly arranged, a small number of goblet cells appeared between the epithelial cells, the intestinal gland in the lamina propria was abundant, and there was no obvious inflammatory cell infiltration. The results are shown in Figure 5A,B.

Compared with the control group, the intestinal tissues of the BRV group showed that the mucosal epithelial cells were cytosolically porous and lightly stained, more mucosal epithelial cells were separated from the lamina propria (green arrow), a small amount of intestinal villi were lost locally (blue arrow), and the structure of the intestinal gland in the lamina propria was unclear. The results are shown in Figure 5C,D.

Compared with the control group, the intestinal surface was a single layer of columnar epithelium, the mucosal epithelial cells were dense, the cells were tightly arranged, cytosolic porosity and light staining were rare, and the goblet cells were scattered among the epithelial cells. A small number of goblet cells appeared between the epithelial cells, and the intestinal glands in the lamina propria were abundant. The results are shown in Figure 5E,F.

## 4. Discussion

There are several screening and analysis methods for intestinal microorganisms. Commonly used molecular biological analysis methods include those based on DNA fingerprinting (ERIC-PCR, DGGE, RAPD) [25,26,27,28], and DNA sequencing technology. In addition, real-time quantitative PCR (RT-qPCR) [29] and fluorescence in situ hybridization (FISH) [30,31] are often used to quantitatively analyze bacteria. Currently, the most common and effective method is high-throughput sequencing [32,33], including whole-genome sequencing and metagenome sequencing. According to the different sequencing principles, purposes, experimental requirements, and species identification depths, different methods can be selected for sequencing.

In this study, 16S rRNA high-throughput sequencing was used to compare the flora of diarrheal and normal calf stools. There were obvious differences in the flora between the two stool samples, especially in relation to the abundance of *Bacteroides*. The abundance of *Bacteroides* in the stools of the normal calves was high, and the abundance of *Bacteroides* in the stools of the diarrheal calves was low. Interestingly, only *Cryptosporidium* was detected in the stools of the normal calves, and the corresponding normal calves had no clinical symptoms of diarrhea. Five pathogens were detected in the stool samples of the calves with diarrhea: *Cryptosporidium*, *E. coli* K99, BCoV, *Giardia*, and BoAstV. Therefore, the abundance of *Bacteroides* affected the number of pathogens causing diarrhea in the calves to a certain extent. The reason may be that *Bacteroides* is rich in outer membrane vesicles (OMVs). *Bacteroides* has been shown to be the main exporter of these OMVs, and *Bacteroides fragilis* and *Bacteroides thetaioatomicron* are the two main players [34,35], The enzymes produced by OMVs can help the intestinal receptor bacteria to decompose a variety of polysaccharides and proteins. More importantly, OMVs can also support the growth of other intestinal flora and contribute to the overall intestinal homeostasis of animals [36,37]. In this study, the appearance of less pathogenic species in normal calf feces and more pathogenic species in diarrhea calf feces may be due to the fact that the abundance of *Bacteroides* supports the growth of other inherent flora, promotes the intestinal homeostasis of calves, and then inhibits the growth of pathogens.

In the previous studies on cattle, only the pathogenic effect of ETBF has been reported [38,39], and no probiotic effect of NTBF has been reported. *Bacteroides* usually plays a beneficial role when colonized in the intestinal tract of animals, but it is a conditional pathogen when colonized in the other parts of animals [40], and *B. fragilis* has shown sufficient beneficial effects in the intestine of humans, chickens, and other animals [18,19,20,21]. In addition to *B. fragilis*, *Bacteroides vulgatus* (*B. vulgatus*), which belongs to Bacteroidaceae, has been shown to prevent animal colitis [41]. *B. vulgatus* promotes the production of succinic acid, propionic acid, and short-chain fatty acids. Short-chain fatty acids, the final metabolites of intestinal microorganisms, can inhibit the growth of pathogenic bacteria by reducing the pH value in the intestine, promoting the proliferation of beneficial bacteria in the intestine, and balancing the intestinal microecological environment. Short-chain fatty acids provide energy sources for intestinal mucosal cells, promote the proliferation of intestinal epithelial cells, help maintain the intestinal barrier, and resist inflammation. *B. vulgatus* can reduce lipopolysaccharide production in the intestine and prevent atherosclerosis [42,43]. It can be seen that *B. vulgatus* indirectly maintains the integrity of the intestinal barrier and plays an important role in the construction of the intestinal immune system. In this study, BRV caused diarrhea symptoms in Kunming mice, which mainly manifested as more separation of the mucosal epithelium and lamina propria in the intestinal tissue, a small number of intestinal villi shedding or shortening, an unclear structure of lamina propria intestinal glands, and other inflammatory reactions. After the intragastric administration of BF-1153 and then the intragastric administration of BRV, the above symptoms can be alleviated, which is manifested by a dense cytoplasm of intestinal mucosal epithelial cells, a tight arrangement between the cells, a rare loose and light staining of the cytoplasm, and an abundant number of intestinal glands in the lamina propria. This is consistent with the report that *B. fragilis* can alleviate intestinal inflammation [44,45], because the interaction between the capsular polysaccharide PSA in *B. fragilis* and the APC cells can enrich the anti-inflammatory memory cells, thereby reducing the occurrence of an inflammatory response [16,17].

In summary, human-derived ETBF has been reported to cause colorectal cancer, whereas animal-derived *B. fragilis* has been reported to have a positive effect on animal-derived colitis, abscesses, and neuroinflammation. Therefore, *B. fragilis* should not generally be attributed to pathogenic bacteria. The role of *B. fragilis* should be viewed dialectically based on the source of the strain, whether it contains enterotoxins, and the amount of bacterial fluid.

## 5. Conclusions

In summary, the strain BF-1153 isolated in this experiment conformed to the growth characteristics of *B. fragilis* on a BBE medium, and did not contain the BFT gene, which proved that it was a strain of NTBF. At the same time, it has no toxic effect on MDBK, HCT-8, and IPEC, and has obvious cell growth-promoting effects. BF-1153 promotes the weight gain of SPF Kunming mice when compared with the control group. At the same time, BF-1153 alleviated the diarrhea symptoms caused by BRV in SPF Kunming mice. In brief, NTBF may be developed into a new generation of probiotics and applied in the animal field.

## Figures and Tables

**Figure 1 microorganisms-12-02319-f001:**
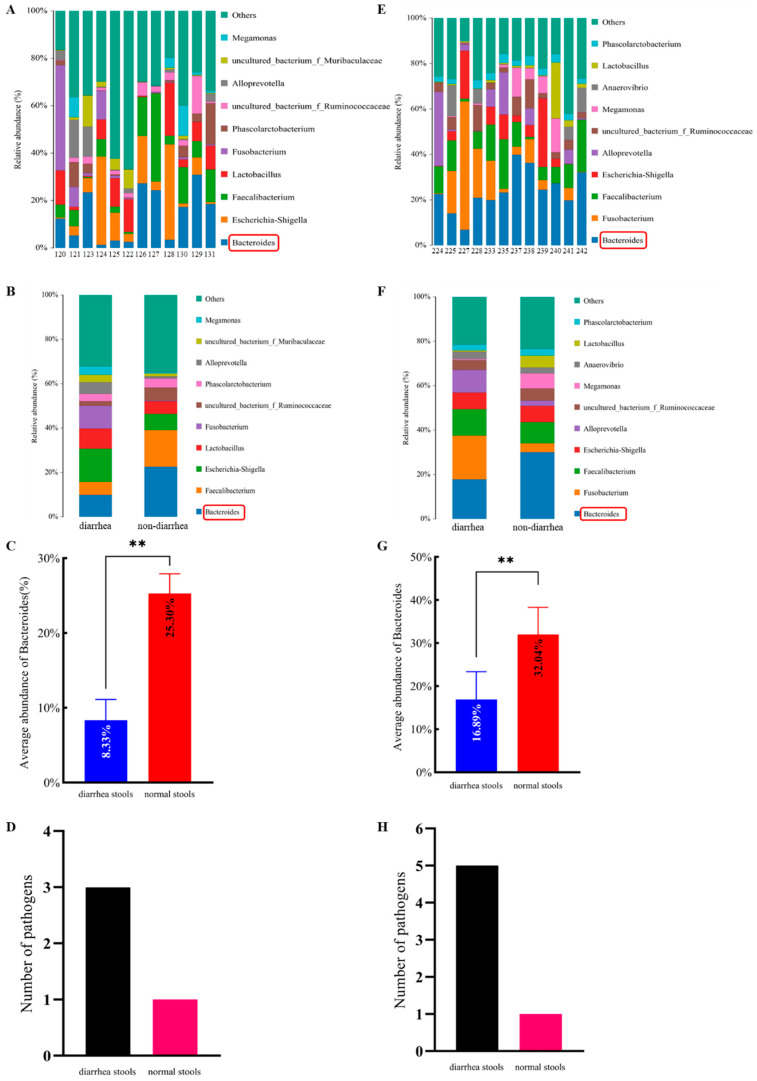
The *Bacteroides* abundance and pathogens for Shangle (**A**–**D**) and Wuli (**E**–**H**) stool samples. The genus-level intestinal microflora map for each sample (**A**,**E**), and the genus-level intestinal microflora map (**B**,**F**), a comparison of the content of *B. fragilis* (**C**,**G**), and the number of pathogens (**D**,**H**) in the diarrheal and normal stools are presented. The data were expressed as the mean ± SD. ** represents extremely significant difference.

**Figure 2 microorganisms-12-02319-f002:**
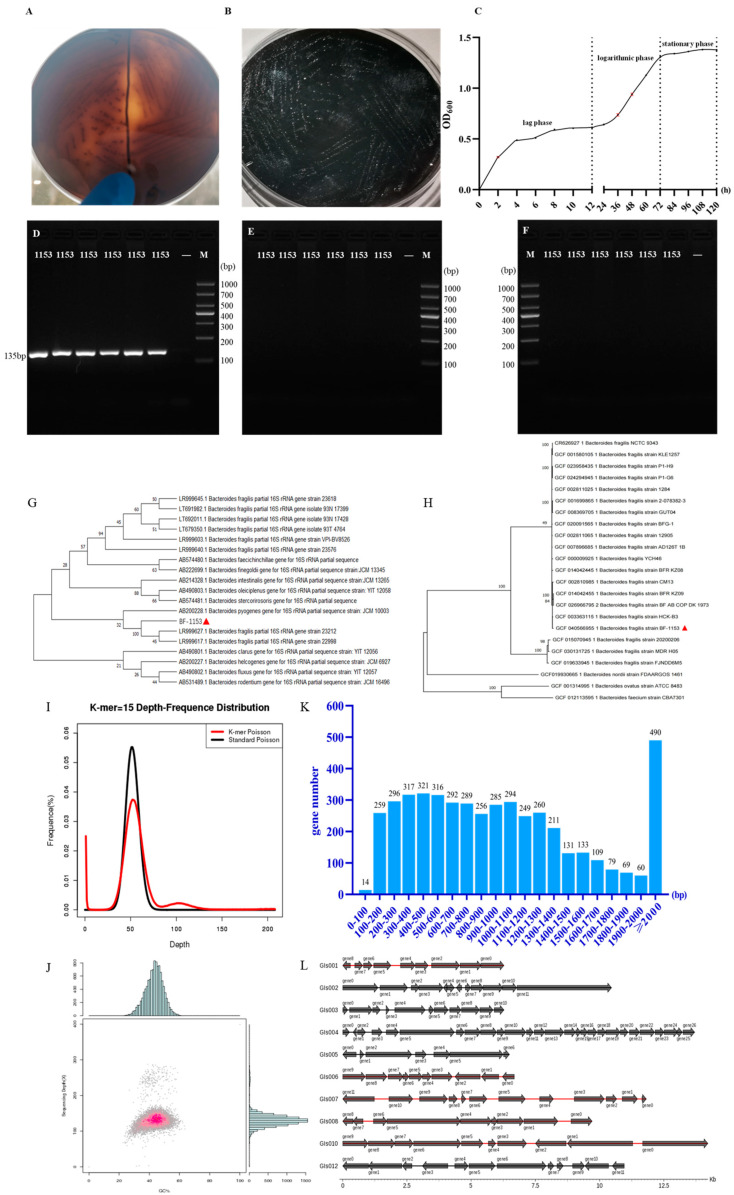
Growth characteristics and genome size of BF-1153 strain. (**A**) BF-1153 grew specific black single colonies on the BBE (front side). (**B**) BF-1153 grew specific black single colonies on the BBE (reverse side). (**C**) The growth curve of BF-1153 (0–12 h is the lag phase, 12–72 h is the logarithmic phase, and 72–120 h is the stationary phase). (**D**) The specific primers to detect BF-1153. (**E**) The detection of BF-1153 by BFT primer. (**F**) The specific primers BFT1, BFT 2, and BFT 3 to detect BF-1153. (**G**) Phylogenetic tree of BF-1153 strain 16S sequence (▲ represents the strain in this study). (**H**) Phylogenetic tree of BF-1153 strain core genome (▲ represents the strain in this study). (**I**) 15-mer statistics of BF-1153 strain. (**J**) BF-1153 GC content and sequencing depth correlation analysis statistics. (**K**) Gene length distribution of BF-1153 strain. (**L**) Gene distribution of GIs in BF-1153 strain.

**Figure 3 microorganisms-12-02319-f003:**
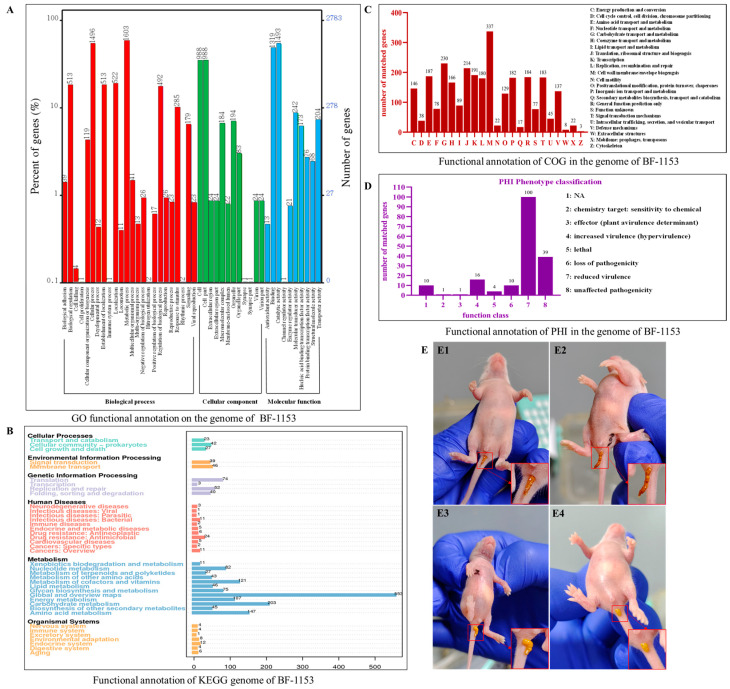
BF-1153 gene function analysis and stool characteristics of SPF Kunming mice treated with different doses of BRV by intragastric irrigation. (**A**) Functional annotation of GO in the genome of BF-1153. (**B**) Functional annotation of KEGG in the genome of BF-1153. (**C**) Functional annotation of COG in the genome of BF-1153. (**D**) Functional annotation of PHI in the genome of BF-1153. (**E1**) After intragastric administration of 100 μL of sterile DMEM, the mice had no diarrhea symptoms, the feces were normally shaped and brown–yellow. (**E2**) After intragastric administration of 25 μL of BRV virus liquid, the feces were shaping, soft, and brown–yellow. (**E3**) After intragastric administration of 50 μL of BRV virus liquid, the feces were unformed, soft, and yellow. (**E4**) After intragastric administration of 100 μL of BRV virus liquid, the feces were unformed, watery, and yellow.

**Figure 4 microorganisms-12-02319-f004:**
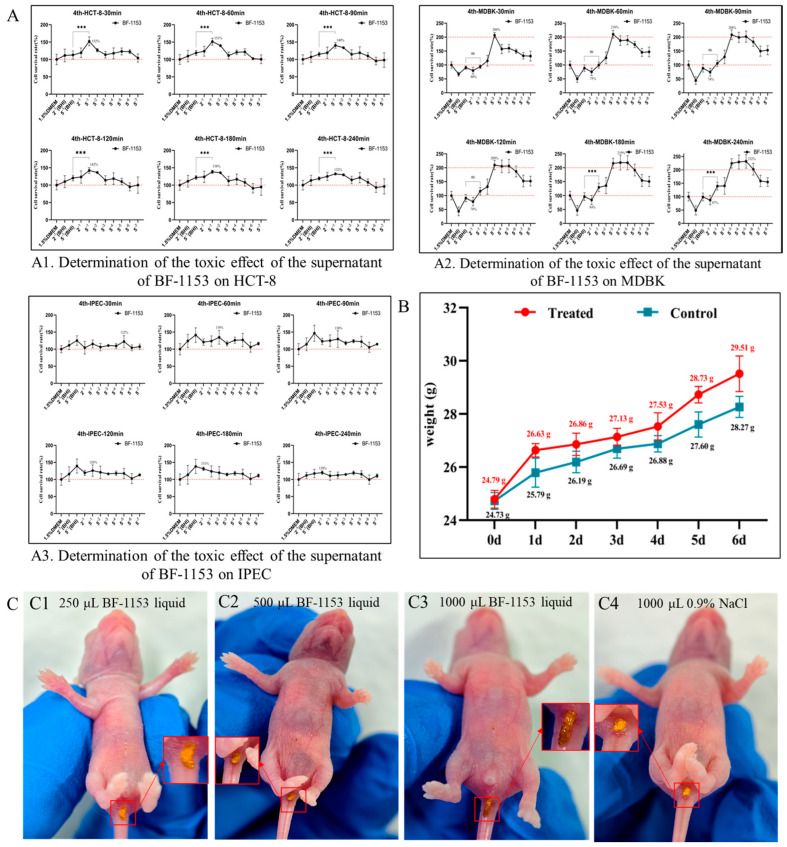
Toxic effect of BF-1153 on healthy SPF Kunming mice and in vitro cells. The data were expressed as the mean ± SD. (**A1**) Determination of the toxic effect of the supernatant of BF-1153 on HCT-8. (**A2**) Determination of the toxic effect of the supernatant of BF-1153 on MDBK. (**A3**) Determination of the toxic effect of the supernatant of BF-1153 on IPEC. (**B**) BF-1153 has a growth-promoting effect on the body weight of healthy mice. (**C1**) Stool characteristics of mice after oral administration of 250 μL of BF-1153 bacterial solution. (**C2**) Stool characteristics of mice after oral administration of 500 μL of BF-1153 bacterial solution. (**C3**) Stool characteristics of mice after oral administration of 1000 μL of BF-1153 bacterial solution. (**C4**) Stool characteristics of mice after oral administration of 1000 μL of 0.9% NaCl solution. *** represents extremely significant difference.

**Figure 5 microorganisms-12-02319-f005:**
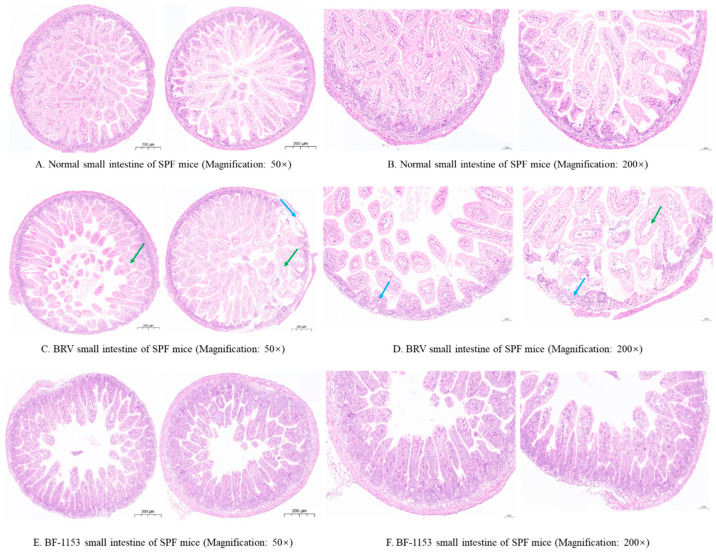
Pathological changes in the small intestine of SPF Kunming mice in BRV group, BF-1153 group, and control group. (**A**) Normal small intestine of SPF mice (Magnification: 50×). (**B**) Normal small intestine of SPF mice (Magnification: 200×). (**C**) BRV small intestine of SPF mice (Magnification: 50×). (**D**) BRV small intestine of SPF mice (Magnification: 200×). (**E**) BF-1153 small intestine of SPF mice (Magnification: 50×). (**F**) BF-1153 small intestine of SPF mice (Magnification: 200×).

**Table 1 microorganisms-12-02319-t001:** Primers used for PCR.

Pathogens Species	Primer	Sequence (5′-3′)	Product Size (bp)	References
16S rRNA	universal primers	27F: AGAGTTTGATCMTGGCTCAG 1492R: GGTTACCTTGTTACGACTT	1500	[22]
*B. fragilis*	LEU	F: CACTTGACTGTTGTAGATAAAGC R: CATCTTCATTGCAGCATTATCC	135	[23]
*B. fragilis*	BFT	F: GGATACATCAGCTGGGTTGTAG R: GCGAACTCGGTTTATGCAGT	296	[24]
*B. fragilis*	BFT1	F: TCTTTTGAATTATCCGTATGCTC R: CTTGGGATAATAAAATCTTAGGGATG	169	[24]
*B. fragilis*	BFT2	F: ATTTTTAGCGATTCTATACATGTTCTC R: GGGCATATATTGGGTGCTAGG	114	[24]
*B. fragilis*	BFT3	F: TGGATCATCCGCATGGTTA R: TTTGGGCATATCTTGGCTCA	148	[24]

**Table 2 microorganisms-12-02319-t002:** The pathogen species and numbers contained in the diarrhea and normal stool from Shangle and Wuli.

Farm	Sample ID of Diarrhea Stools	Types of Pathogens	Sample ID of Normal Stools	Types of Pathogens
Shangle	121	*E.coli* K99 (1)	120	*Cryptosporidium* (1)
	122	*Cryptosporidium* (1)	123	-
	124	*Cryptosporidium* (1)	126	-
	125	*Cryptosporidium* (1)	127	-
	128	*Cryptosporidium* (1)	129	*Cryptosporidium* (1)
	130	BCoV, *Cryptosporidium* (2)	131	-
	total	3	total	1
Wuli	224	*Cryptosporidium*, *Giardia* (2)	237	*Cryptosporidium* (1)
	225	*Cryptosporidium*, BoAstV (2)	238	-
	227	BCoV, *Cryptosporidium* (2)	239	-
	228	*Cryptosporidium* (1)	240	-
	233	*E.coli* K99, *Cryptosporidium*, *Giardia* (3)	241	-
	235	*Cryptosporidium* (1)	242	*Cryptosporidium* (1)
	total	5	total	1

**Table 3 microorganisms-12-02319-t003:** Sequence alignment results for BF-1153 strain with other *Bacteroides fragilis* strains.

Strains Name	Description	Query Cover	Percent Identity	Accession
*B. fragilis* NCTC 9343	16S ribosomal RNA	98%	99.51%	NR_074784.2
*B. fragilis* ATCC 25285	16S ribosomal RNA	98%	99.51%	NR_119164.1
*B. fragilis* JCM 11019	16S ribosomal RNA	98%	99.51%	NR_112936.1
*B. fragilis* DSM 2151	16S ribosomal RNA	95%	98.11%	NR_112936.1

**Table 4 microorganisms-12-02319-t004:** Virulence factors predicted in VFDB.

Gene ID	VF-ID	VF Name	Related Genes
BF-1153-GM001508	VF0504	AdeFGH efflux pump	*adeG*
BF-1153-GM001779	VF0003	Capsule	*cap8J*
BF-1153-GM002190	VF0465	Capsule	ACICU_00080
BF-1153-GM002201	VF0465	Capsule	ACICU_00076
BF-1153-GM001522	VF0072	ClpC	*clpC*
BF-1153-GM003898	VF0074	ClpP	*clpP*
BF-1153-GM000212	VF0326	LOS	*Cj1135*
BF-1153-GM000761	VF0171	LPS	*yvfE*
BF-1153-GM003443	VF0367	LPS	*gmd*
BF-1153-GM004056	VF0542	LPS	*wbtI*
BF-1153-GM004650	VF0171	LPS	*hisF*
BF-1153-GM002680	VF0153	Mip	*mip*
BF-1153-GM000735	VF0392	O-antigen	*ddhA*
BF-1153-GM004424	VF0473	Polar flagella	*flmH*
BF-1153-GM003812	VF0414	RicA	*ricA*
BF-1153-GM004103	VF0169	SodB	*sodB*

**Table 5 microorganisms-12-02319-t005:** Predicted resistance genes in ARDB and CARD.

Gene ID	Identity	Resistance Type	Original Resistance Type	Drug Class
BF-1153-GM001442	99.5%	*mefA*	mls_mfs	Lincosamide; Macrolide; Oxazolidinone; Phenicol; Pleuromutilin; Streptogramin; Tetracycline
BF-1153-GM003581	99.7%	*bl2e_cepA*	bla_a	Cephalosporin
BF-1153-GM003998	99.6%	*ermF*	erm	Lincosamide; Macrolide; Streptogramin
BF-1153-GM004000	99.5%	*mefA*	mls_mfs	Lincosamide; Macrolide; Oxazolidinone; Phenicol; Pleuromutilin; Streptogramin; Tetracycline
BF-1153-GM004302	98.8%	*tetQ*	tet_rpp	Tetracycline
BF-1153-GM004304	99.6%	*ermF*	erm	Lincosamide; Macrolide; Streptogramin

**Table 6 microorganisms-12-02319-t006:** The drug resistance phenotype of BF-1153 strain.

Antibiotics Name	Determination of Drug Sensitivity (mm)	Diameter of Drug Sensitive Tablets of BBE (mm)
lincomycin	≥21	6
erythromycin	≥23	6
azithromycin	≥18	6
tetracycline	≥19	6
doxycycline	≥16	6
trimethoprim	≥16	6

Note: The antimicrobial susceptibility of BF-1153 against 6 antimicrobial agents (Hangzhou Microbial Reagents Co., Ltd., Hangzhou, China) was valued by the disc diffusion method according to the recommendation of the Clinical and Laboratory Standards Institute (CLSI, 2020) (https://clsi.org/, accessed on 23 September 2023).

## Data Availability

The data that support the findings of this study are available from the corresponding author upon reasonable request. The whole-genome sequencing results for the BF-1153 strain have been uploaded to NCBI. BioProject: PRJNA1096115, BioSample: SAMN40744850, Genomes: JBEXBY000000000.

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
