# Peer review of "Analysis of Characteristics of Bovine-Derived Non-Enterotoxigenic Bacteroides fragilis and Validation of Potential Probiotic Effects"

_microorganisms, 2024, doi:10.3390/microorganisms12112319_

Round 1

Reviewer 1 Report

Comments and Suggestions for Authors

In the present manuscript, the authors have Using 16S rRNA high-throughput sequencing, we determined the microbial composition of diarrheal and normal stools by 16S rRNA high-throughput sequencing and explored the differences in intestinal microorganisms between calves with diarrhea and normal calves. They have also isolated and cultured the key intestinal microorganisms and verified their biological characteristics.This study showed that that a bovine-derived Bacteroides fragilis strain can enhance the activities of HCT-8, IPEC, and MDBK cells, especially bovine-derived MDBK. Animal experiments showed that BF-1153 had no toxic effects on SPF Kunming  mice, and BRV-induced diarrhea in SPF Kunming mice had a certain relief effect. 

This is a worth published study and these findings provide new insights on new generation probiotics, such as B. fragilis. The methodology used is sound, nonetheless some iprovements should be made considering mostly the presentation of the results. A moderate editing in Enlish is also required before publication. 

Major comments:

- line 17: please clarify the ''large-scale''. Do you mean the number of animals? Please provide this information in the methods, section ''sampling''. 

- line 18: please correct as ''13 diarrheal and 11 normal stools'' and check the numbers throughout the manucscript and Table S1. There are 12 diarrheal and 12 normal stools presented in Table 2. Moreover, state and clarify how many samples were positive by PCR, using specific primers for identification of B. fragilis, name here strain BF-1153 and delete lines 26-27 for consistency. Was BF-1153 the only strain recovered from these samples? If not, why this strain was chosen for further analysis. These points also need to be clarified in the results section (lines 128-129) in more detail for clarity. 

- In the introduction, please provide more information on diarhoea (please see at https://www.who.int/news-room/fact-sheets/detail/diarrhoeal-disease), e.g, definition, mortality, causes. 

- lines 94-95 describe the aims of the study, and should be moved to lines 98-99. Please rephrase for clarity such as: ''Using 16S rRNA high-throughput sequencing, we determined the microbial composition of diarrheal and normal stools and explored the differences in intestinal microorganisms between calves with diarrhea and normal calves. Finally, we isolated and cultured the key intestinal microorganisms and verified their biological characteristics.''

- The presentation of the figures and tables should be improved. Table 3 may be presented as a supplementary Table. The data presented in the Figures is also excessive and the fonts need to be enlarged.The photographs from the animal experiments (Figures 3E1-E4 and Figures 4C1-C4) and Figures 2H-I may be presented in the supplementary material. Please rotate Figures 3A-D and Figures 4A, 4B. 

- The figure legends may be more condense, e.g.

Figure 1. The Bacteroides abundance and pathogens in Shangle (A-D) and Wuli (E-H). The genus-level intestinal microflora map for each sample (A, E), and the genus-level intestinal microflora map (B, F), comparison of the content of B. fragilis (C, G), and the number of pathogens (D, H) between diarrheal and normal stools are presented. The data were expressed as the mean±SD.

- lines 29-30 and 43-44 provide the same information, thus please delete lines 43-44 and clarify the statement: ''belonged to the same branch as B. fragilis strains 23212 and 22998 and other common Bacteroides'' providing more information about these two strains (e.g. country of origin, source and year of isolation), and in more detail in the Methods section (lines 302-303) and explaining what do you mean ''common Bacteroides''?

- A phylogenomic analysis, e.g SNP-based tree or core-genome MLST (cgMLST) analysis should also be performed to compare the WGS assembly of strain BF-1153 with other previously published B. fragilis genomes.

- lines 329-360: please correct as ''Component analysis of bovine-derived BF-1153 genome'' and remove the titles of the subsections. These data may be presented in a single paragraph. 

- Table 6: please define the size diameters in mm and add a footnote on the criteria used for the breakpoints used for drug sensitivity/resistance (EUCAST, CLSI).

- In the coclusions, more information from the results of this study should be presented.

Minor comments:

- Please write as ''Bacteroides fragilis'' in first occurence and then B. fragilis in italics throughout the text (line 15: please delete B. fragilis and the parenthesis).

- please provide the full names of the viruses (BRV, BoAstV, and BCoV

- line 19 and 101: please replace ''screened'' with ''determined''. 

- lines 20-21: please correct as: ''Genome composition and in silico genome functional analyses were carried out to reflect the biological characteristics of BF-1153 strain''. 

- line 34: please clarify what do you mean with '''changes''. Are they muatations in antimicrobial and virulence genes?''

- line 26: please define the ''CCK8'' as: ''Cell Counting Kit-8 (CCK8')'' and clarify ''the strain on the related cells'' as: ''the BF-1153 stain on three different cell lines; human ileocecal colorectal adenocarcinoma (HCT-8) cells, Madin-Darby bovine kidney (MDBK) cells and intestinal porcine epithelial cells  (IPEC).''

-line 103: please change the title such as ''sampling and 16S high-throughput sequencing'', delete line 108 and add ''to determine the key microorganisms in the gut'' in the end of the sentence in line 112.

- line 106: please use plural for ''sample'' as ''The details of the samples...''

-line 117: please replace the coma with the point symmbol such as ''...medium.''

- line 138: please rephrase ''The DNA of the positive B. fragilis strain identified using specific primers, was extracted....'' as'' DNA from strain BF-1153, which was positive by PCR using the B. fragilis specific primers, was extracted....''

-line 144: please replace ''Genome sequencing analysis of BF-1153 genome components'' with ''Whole-genome sequencing (WGS) analysis of BF-1153 strain''.

-line 159: please replace ''strains'' with ''BF-1153 strain''.

-line 188: please delete on cells.

- line 171: please replace ''Whole genome sequencing analysis of BF-1153 genome function'' with ''Functional annotation of the of BF-1153 genome''.

-line 178: please correct as: ''functional annotation obtained by ARDB....''.

-line 188: please replace ''identification'' with ''investigation''

- line 195: please replace ''HCT-8 cells'' with ''Each of the three cell lines (HCT-8, MDBK, and IPEC)...'' and delete ''The MDBK and IPEC cells repeated this operation'' in line 201.

- line 199: please replace ''observation'' with ''incubation'' and clarify as  ''the CCK-8 Cell Proliferation and Toxicity Detection Kit...''

- line 207: please define DMEM.

-line 211: please define the CFUs (i.e. colony-forming units)

- line 222: please correct as ''diarrhoea''

- line 242: please use plural such as ''sequences were obtained''.

- lines 277-278: Please denote that the numders denot the sample IDs, not the no. of samples in the titles of columns 2, 4 and present in parentheses the numer f isolates instead the types of pathogenes in colummns 3,5 such as: types of pathogens (no. of isolates), e.g. E. coli K99 (1).

-lines 284-285 may be deleted as this information is also presented in the legend of figure 2 (lines 307-308).

- line 460: please correct as ''Investigation of toxic effects of bovine-derived BF-1153 on cell lines''

- line 474:  please correct as ''Investigation of toxic effects of bovine-derived BF-1153 on SPF Kunming mice''.

- line 542: please replace ''diarrheal calf stools and normal calf stools'' with '' diarrheal  and normal calf stools''

-line 546: please delete '''as shown in Fig 1A and 1E''. 

-line 551-553 : please rephrase for clarity.

- line 562: please replace ''In study,'' with ''In previous studies'' 

- lines 579-582: please delete (Fig.5C, 5D), (green arrows), (blue arrows)

- line 587: please delete (Fig.5E, 5F).

Comments on the Quality of English Language

Moderate editing in English is required

Author Response

Major comment:

Comment 1:  line 17: please clarify the ''large-scale''. Do you mean the number of animals? Please provide this information in the methods, section ''sampling''. 

Response 1: Thank you very much for your valuable suggestion. 'Large-scale ' refers to a centralized, standardized, non-free-range cattle farm, not the number of animals.

Comment 2:  line 18: please correct as ''13 diarrheal and 11 normal stools'' and check the numbers throughout the manuscript and Table S1. There are 12 diarrheal and 12 normal stools presented in Table 2. Moreover, state and clarify how many samples were positive by PCR, using specific primers for identification of B. fragilis, name here strain BF-1153 and delete lines 26-27 for consistency. Was BF-1153 the only strain recovered from these samples? If not, why this strain was chosen for further analysis. These points also need to be clarified in the results section (lines 128-129) in more detail for clarity. 

Response 2: Thank you very much for your valuable suggestion. It has been corrected to '12 diarrheal and 12 normal stools' in the manuscript, and table 2 and table S1 have been modified. BF-1153 is the only strain in these samples, and other problems have been explained in 'Growth characteristics of bovine-derived Bacteroides fragilis'.

Comment 3: In the introduction, please provide more information on diarhoea (please see at https://www.who.int/news-room/fact-sheets/detail/diarrhoeal-disease), e.g, definition, mortality, causes. 

Response 3: Thank you very much for your valuable suggestion. It has been modified in the introduction。

Comment 4:  lines 94-95 describe the aims of the study, and should be moved to lines 98-99. Please rephrase for clarity such as: ''Using 16S rRNA high-throughput sequencing, we determined the microbial composition of diarrheal and normal stools and explored the differences in intestinal microorganisms between calves with diarrhea and normal calves. Finally, we isolated and cultured the key intestinal microorganisms and verified their biological characteristics.''

Response 4: Thank you very much for your valuable suggestion.  It has been revised and rephrased.

Comment 5: The presentation of the figures and tables should be improved. Table 3 may be presented as a supplementary Table. The data presented in the Figures is also excessive and the fonts need to be enlarged. The photographs from the animal experiments (Figures 3E1-E4 and Figures 4C1-C4) and Figures 2H-I may be presented in the supplementary material. Please rotate Figures 3A-D and Figures 4A, 4B. 

Response 5: Thank you very much for your valuable suggestion. It has partially adjusted the presentation of the figures.

Comment 6: The figure legends may be more condense, e.g. Figure 1. The Bacteroides abundance and pathogens in Shangle (A-D) and Wuli (E-H). The genus-level intestinal microflora map for each sample (A, E), and the genus-level intestinal microflora map (B, F), comparison of the content of B. fragilis (C, G), and the number of pathogens (D, H) between diarrheal and normal stools are presented. The data were expressed as the mean ± SD.

Response 6: Thank you very much for your valuable suggestion. It has been revised.

Comment 7:  lines 29-30 and 43-44 provide the same information, thus please delete lines 43-44 and clarify the statement: ''belonged to the same branch as B. fragilis strains 23212 and 22998 and other common Bacteroides'' providing more information about these two strains (e.g. country of origin, source and year of isolation), and in more detail in the Methods section (lines 302-303) and explaining what do you mean ''common Bacteroides''?

Response 7: Thank you very much for your valuable suggestion. "other common Bacteroides" has been deleted, and other parts have been adjusted and modified.

Comment 8: A phylogenomic analysis, e.g SNP-based tree or core-genome MLST (cgMLST) analysis should also be performed to compare the WGS assembly of strain BF-1153 with other previously published B. fragilis genomes.

Response 8: Thank you very much for your valuable suggestion. Since the genome size of BF-1153 is around 5.71 Mb, there are fewer B. fragilis genomes of the same size in the NCBI database, and thus 16S sequences were considered for cluster analysis.

Comment 9: lines 329-360: please correct as ''Component analysis of bovine-derived BF-1153 genome'' and remove the titles of the subsections. These data may be presented in a single paragraph. 

Response 9: Thank you very much for your valuable suggestion. It has been revised and rephrased.

Comment 10: Table 6: please define the size diameters in mm and add a footnote on the criteria used for the breakpoints used for drug sensitivity/resistance (EUCAST, CLSI).

Response 10: Thank you very much for your valuable suggestion. Footnotes have been added to Table 6 for clarification.

Comment 11: In the coclusions, more information from the results of this study should be presented.

Response 11: Thank you very much for your valuable suggestion. It has been revised and rephrased.

Comment 12: Minor comments

Response 12: Other "Minor comments" have been corrected and adjusted in the manuscript.

Reviewer 2 Report

Comments and Suggestions for Authors

The authors did an amazing job drafting the manuscript. However, there are some suggestions that can enhance the quality of this research manuscript, as outlined below:

1. Line 15-30:
The abstract is quite lengthy, and repetition is observed in the introduction. The authors are suggested to condense the text and focus only on the core ideas in the abstract.

2. Line 31-32:
"The results showed that BF-1153 had no toxic effects on HCT-8, MDBK, and IPEC. Animal experiments have shown that BF-1153 has no toxic effects on healthy SPF Kunming mice."

The authors are suggested to include the key results with quantitative values and standard units. In the statement "no toxic results," please provide key values for comparison with diarrheal samples. The inclusion of values will make the abstract more precise.

3. Line 52-53:
Introduction: "Diarrhea is one of the most common safety concerns worldwide, seriously affecting both people and animals."

The authors are suggested to add the most recent mortality rates caused by diarrhea in both humans and animals, citing updated references.

4. The authors are recommended to present the results in a schematic diagram showing the clinical findings along with the methodology and experimental design.

Figure: Schematic illustration of the characteristics of a bovine-derived non-enterotoxigenic Bacteroides fragilis and validation of potential probiotic effects

Author Response

Comment 1:  Line 15-30: The abstract is quite lengthy, and repetition is observed in the introduction. The authors are suggested to condense the text and focus only on the core ideas in the abstract.

Response 1: Thank you very much for your valuable suggestion. “Abstract” has been revised.

Comment 2:  Line 31-32: "The results showed that BF-1153 had no toxic effects on HCT-8, MDBK, and IPEC. Animal experiments have shown that BF-1153 has no toxic effects on healthy SPF Kunming mice." The authors are suggested to include the key results with quantitative values and standard units. In the statement "no toxic results," please provide key values for comparison with diarrheal samples. The inclusion of values will make the abstract more precise.

Response 2: Thank you very much for your valuable suggestion. BF-1153 was isolated from healthy calves (calves without diarrhea symptoms), and then its toxicity was identified by cell line toxicity identification and animal experiment toxicity identification, which did not involve comparison with diarrhea samples.

Comment 3: Line 52-53: Introduction: "Diarrhea is one of the most common safety concerns worldwide, seriously affecting both people and animals." The authors are suggested to add the most recent mortality rates caused by diarrhea in both humans and animals, citing updated references.

Response 3: Thank you very much for your valuable suggestion. Introduction has been modified.

Comment 4:  The authors are recommended to present the results in a schematic diagram showing the clinical findings along with the methodology and experimental design.

Response 4: Thank you very much for your valuable suggestion. The schematic diagram has been uploaded separately. 

Reviewer 3 Report

Comments and Suggestions for Authors

The manuscript is hard to read and understand. The abstract is too long; it must be written more concisely and ...more short.

1)Before the chapter entitled ''Materials and methods,'' authors must write the objectives of their studies.

2)The methodology presented at the beginning contains mistakes; this part must be rewritten by the microbiologist who performed the tests.

3)The ethics concept regarding the research performed on  animals is missing; 

4)Here, the authors must explain why they use a model on mice to demonstrate the nonharmful nature of some bacteria isolated from cattle. The digestive system of cattle is different from the digestive system of mice. 

5) The HCT-8 cell line must be introduced in methodology; authors must write the provenience of it and the reason for which they have used this cell line in their research;

6) The results must be presented more concisely;

7) More pictures  from the manuscript are not clear; authors must supply here pictures of high quality;

8) In the text of the manuscript, the bibliographic references are not written according to the MDPI style

9)The English language used in this article is not proper. 

10)More sentences must be rewritten because they cannot be understood.

Comments on the Quality of English Language

The English language used in this article is not proper. 

10)More sentences must be rewritten because they cannot be understood.

Author Response

Comment 1:Before the chapter entitled ''Materials and methods,'' authors must write the objectives of their studies.

Response 1:Thank you very much for your valuable suggestion. Changes have been made in the last paragraph of the "Introduction".

Comment 2: The methodology presented at the beginning contains mistakes; this part must be rewritten by the microbiologist who performed the tests.

Response 2: Thank you very much for your valuable suggestion. Please point out the specific errors in order to modify and adjust.

Comment 3: The ethics concept regarding the research performed on animals is missing.

Response 3: Thank you very much for your valuable suggestion. Animal experiments were carried out under the supervision of the Executive Committee of Laboratory Animal Management and Ethics Inspection of the Northwest A&F University. Animal experiments were in full compliance with ethical requirements and have been explained in the ' Ethics statement ' section of the manuscript.

Comment 4: Here, the authors must explain why they use a model on mice to demonstrate the nonharmful nature of some bacteria isolated from cattle. The digestive system of cattle is different from the digestive system of mice. 

Response 4: Thank you very much for your valuable suggestion. BF-1153 was isolated from healthy calves, i.e. calves without diarrhea symptoms, SPF Kunming mice are standard and cost-effective experimental animals, the difference between the digestive systems of cattle and mice mainly lies in the number of stomachs, but the correlation between BRV infection experiments and BF-1153 toxicity assay experiments with stomachs in this experiment is low (BRV infection mainly leads to intestinal symptoms, and B. fragilis is mainly set in the colon), so SPF Kunming mice were chosen for animal experiments.

Comment 5: The HCT-8 cell line must be introduced in methodology; authors must write the provenience of it and the reason for which they have used this cell line in their research.

Response 5: Thank you very much for your valuable suggestion. All cells in the manuscript were preserved and provided by our laboratory.HCT-8 was derived from human colon. To investigate the toxic response of BF-1153 to intestinal cells, intestinal cells of human and porcine origin and MDBK cells of bovine origin were selected separately.

Comment 6: The results must be presented more concisely.

Response 6: Thank you very much for your valuable suggestion. The 'Component analysis of bovine-derived BF-1153 genome' section of the Result has been adjusted.

Comment 7: More pictures from the manuscript are not clear; authors must supply here pictures of high quality.

Response 7: Thank you very much for your valuable suggestion. The clarity of the pictures in the manuscript has been adjusted and all the pictures have been uploaded separately.

Comment 8: The English language used in this article is not proper. 

Response 8: Thank you very much for your valuable suggestion. Some English language has been adjusted.

Comment 9: More sentences must be rewritten because they cannot be understood.

Response 9: Thank you very much for your valuable suggestion. Some English language has been adjusted.

Round 2

Reviewer 1 Report

Comments and Suggestions for Authors

The authors have successfully replied to most of the reviewers comments. The phylogenetic analysis using the 16sRNA gene is not as accurate as using  a core genome  analysis to compare the BF-1153  strain with other published B. fragilis genomes in the NCBI database (please see at https://www.ncbi.nlm.nih.gov/datasets/genome/?taxon=817; Valdezate, S.; Cobo, F.; Monzón, S.; Medina-Pascual, M.J.; Zaballos, Á.; Cuesta, I.; Pino-Rosa, S.; Villalón, P. Genomic Background and Phylogeny of cfiA-Positive Bacteroides fragilis Strains Resistant to Meropenem-EDTA. Antibiotics 2021, 10, 304. https://doi.org/10.3390/antibiotics10030304) or a Multilocus Sequence Analysis to compare the BF-1153 strain with other isolates in the PubMLST database   (please see at: https://pubmlst.org/bigsdb?db=pubmlst_bfragilis_isolates&page=query; Nielsen FD, Skov MN, Sydenham TV, Justesen US. Development and Clinical Application of a Multilocus Sequence Typing Scheme for Bacteroides fragilis Based on Whole-Genome Sequencing Data. Microbiol Spectr. 2023 Mar 21;11(2):e0511122. doi: 10.1128/spectrum.05111-22. Epub ahead of print. PMID: 36943061; PMCID: PMC10101032.). Please make the appropriate amendments 

Author Response

Comment 1:The authors have successfully replied to most of the reviewers comments. The phylogenetic analysis using the 16sRNA gene is not as accurate as using  a core genome  analysis to compare the BF-1153  strain with other published B. fragilis genomes in the NCBI database (please see at https://www.ncbi.nlm.nih.gov/datasets/genome/?taxon=817; Valdezate, S.; Cobo, F.; Monzón, S.; Medina-Pascual, M.J.; Zaballos, Á.; Cuesta, I.; Pino-Rosa, S.; Villalón, P. Genomic Background and Phylogeny of cfiA-Positive Bacteroides fragilis Strains Resistant to Meropenem-EDTA. Antibiotics 2021, 10, 304. https://doi.org/10.3390/antibiotics10030304) or a Multilocus Sequence Analysis to compare the BF-1153 strain with other isolates in the PubMLST database   (please see at: https://pubmlst.org/bigsdb?db=pubmlst_bfragilis_isolates&page=query; Nielsen FD, Skov MN, Sydenham TV, Justesen US. Development and Clinical Application of a Multilocus Sequence Typing Scheme for Bacteroides fragilis Based on Whole-Genome Sequencing Data. Microbiol Spectr. 2023 Mar 21;11(2):e0511122. doi: 10.1128/spectrum.05111-22. Epub ahead of print. PMID: 36943061; PMCID: PMC10101032.). Please make the appropriate amendments 

Response: Thank you very much for your valuable suggestion. A core genome analysis has been added to the manuscript and an evolutionary tree constructed.

Reviewer 3 Report

Comments and Suggestions for Authors

Comment 2: The methodology presented at the beginning contains mistakes; this part must be rewritten by the microbiologist who performed the tests.

Authors wrote in their manuscript: ‘’ After sending the collected diarrheal stool samples and normal stool samples to the  laboratory at 4°C, the fecal samples were aliquoted in 200g to 2 mL sterile EP tubes…’’

In which way is it possible to introduce 200 g from any material in an Eppendorf tube of 2 mL? 

Comment 4: Here, the authors must explain why they use a model on mice to demonstrate the nonharmful nature of some bacteria isolated from cattle. The digestive system of cattle is different from the digestive system of mice. 

The author's response must be introduced in the text of the manuscript in a scientific manner, not as an answer for the reviewer (sentences sustained with bibliographic references). The readers must understand who are the limitations of the model performed on mice and what are the scientific relevance of the experiments performed on mice in this case. Otherwise, the tests made on mice must be removed from the manuscript

Comment 5: The HCT-8 cell line must be introduced in methodology; authors must write the provenience of it and the reason for which they have used this cell line in their research.

b)The provenience of each standardised cell line used must be written according to the rules established by the commercial supplier. It is not sufficient to write’’ was purchased from ATCC’’. If all cell lines were supplied by ATCC, then these must be written in the manuscript according to the rules of ATCC.

c). Clarify in the manuscript the relevance of using the human cell line in a scientific manner,  with bibliographic references; otherwise, the tests performed on the human cells must be removed from the manuscript.

Comment 7: More pictures from the manuscript are not clear; authors must supply here pictures of high quality, respectively Figure 2K;      Figures 3 A,C, B. 

Author Response

Comment 2: The methodology presented at the beginning contains mistakes; this part must be rewritten by the microbiologist who performed the tests.

Authors wrote in their manuscript: ‘’ After sending the collected diarrheal stool samples and normal stool samples to the  laboratory at 4°C, the fecal samples were aliquoted in 200g to 2 mL sterile EP tubes…’’

In which way is it possible to introduce 200 g from any material in an Eppendorf tube of 2 mL? 

Response 2: Thank you very much for your valuable suggestion.  This sentence means “Dispense feces into multiple EP tubes rather than one EP tube”. Changes have been made in the manuscript.

Comment 4: Here, the authors must explain why they use a model on mice to demonstrate the nonharmful nature of some bacteria isolated from cattle. The digestive system of cattle is different from the digestive system of mice. 

The author's response must be introduced in the text of the manuscript in a scientific manner, not as an answer for the reviewer (sentences sustained with bibliographic references). The readers must understand who are the limitations of the model performed on mice and what are the scientific relevance of the experiments performed on mice in this case. Otherwise, the tests made on mice must be removed from the manuscript.

Response 4: Thank you very much for your valuable suggestion.  The use of exogenous B. fragilis strains for cross-species correlation studies has been very common, e.g., the study of B. fragilis NCTC 9343 on mouse kidney (Zhou, W., Wu, W. H., Si, Z. L., Liu, H. L., Wang, H., Jiang, H., Liu, Y. F., Alolga, R. N., Chen, C., Liu, S. J., Bian, X. Y., Shan, J. J., Li, J., Tan, N. H., & Zhang, Z. H. (2022). The gut microbe Bacteroides fragilis ameliorates renal fibrosis in mice. Nature communications, 13(1), 6081. https://doi.org/10.1038/s41467-022-33824-6 ) and the use of experimental mice for virulence characterization of exogenous B. fragilis strains (Liu Y, Zhang W, Bai Y, Zhi F. [Isolation and identification of a non-enterotoxigenic strain of Bacteroides fragilis from a healthy term infant]. Zhonghua Yi Xue Za Zhi. 2014 Aug 13;94(30):2372-4. Chinese. PMID: 25399982.) have fully demonstrated that experimental mice can be used for cross-species pathogen correlation studies.

Comment 5: The HCT-8 cell line must be introduced in methodology; authors must write the provenience of it and the reason for which they have used this cell line in their research.

b)The provenience of each standardised cell line used must be written according to the rules established by the commercial supplier. It is not sufficient to write’’ was purchased from ATCC’’. If all cell lines were supplied by ATCC, then these must be written in the manuscript according to the rules of ATCC.

c). Clarify in the manuscript the relevance of using the human cell line in a scientific manner,  with bibliographic references; otherwise, the tests performed on the human cells must be removed from the manuscript.

Response 5: Thank you very much for your valuable suggestion.  HCT-8 cells have been used many times to isolate or culture pathogens from cattle such as bovine coronavirus (Jiménez-Meléndez A, Shakya R, Markussen T, Robertson LJ, Myrmel M, Makvandi-Nejad S. Gene expression profile of HCT-8 cells following single or co-infections with Cryptosporidium parvum and bovine coronavirus. Sci Rep. 2023 Dec 13;13(1):22106. doi: 10.1038/s41598-023-49488-1. PMID: 38092824; PMCID: PMC10719361.)(Wu, S., Shao, T., Xie, J., Li, J., Sun, L., Zhang, Y., Zhao, L., Wang, L., Li, X., Zhang, L., & Wang, R. (2024). MiR-199a-3p regulates HCT-8 cell autophagy and apoptosis in response to Cryptosporidium parvum infection by targeting MTOR. Communications biology, 7(1), 924. https://doi.org/10.1038/s42003-024-06632-5), which fully demonstrates that HCT-8 cells have been widely used in the study of pathogens of animal origin.

Comment 7: More pictures from the manuscript are not clear; authors must supply here pictures of high quality, respectively Figure 2K;      Figures 3 A,C, B. 

Response 7: Thank you very much for your valuable suggestion.  PDF version of the picture is compressed, please view the submitted word version.